# Personal and General Views on Aging, Non-Communicable Diseases, and Their Interaction as Cross-Sectional Correlates of Vigorous Physical Activity in UK Individuals Aged 50+

**DOI:** 10.3390/healthcare13162071

**Published:** 2025-08-21

**Authors:** Fabrizio Mezza, Daniela Lemmo, Maria Francesca Freda, Victoria Tischler, Blossom C. M. Stephan, Maria Mataró, Serena Sabatini

**Affiliations:** 1Department of Humanities, University of Naples Federico II, 80131 Naples, Italy; fabrizio.mezza@unina.it (F.M.); daniela.lemmo@unina.it (D.L.); mariafrancesca.freda@unina.it (M.F.F.); 2School of Psychology, Faculty of Health and Medical Sciences, University of Surrey, Guildford GU2 7XH, UK; v.tischler@surrey.ac.uk; 3Dementia Centre of Excellence, EnAble Institute, Faculty of Health Sciences, Curtin University, Perth, WA 6102, Australia; blossom.stephan@curtin.edu.au; 4Institut de Neurociències, Universitat de Barcelona, 08028 Barcelona, Spain; mmataro@ub.edu; 5Institut de Recerca Sant Joan de Déu. Esplugues de Llobregat, 08028 Barcelona, Spain; 6School of Clinical Psychology and Psychobiology, University of Barcelona, 08028 Barcelona, Spain

**Keywords:** awareness of age-related change, expectations regarding aging, self-perceptions of aging, multimorbidity, comorbidity, older adults

## Abstract

**Background**: This study investigated the cross-sectional associations of personal and general views on aging, number of non-communicable diseases, and their interactions as cross-sectional predictors of vigorous physical activity. **Methods**: Participants were 1699 individuals aged 50 years and over (Mean age = 67.79) and living in the community in the UK; 70.8% were women. Participants completed measures assessing Awareness of Age-Related Gains and Losses (AARC-Gains; AARC-Losses; indicators of personal views on aging), Expectations Regarding Aging (ERA; indicator of general views on aging), vigorous physical activity in the last month, non-communicable disease status, and sociodemographic questions. Linear regression models were used. **Results**: After having adjusted for age, sex, education, marital status, and working status, higher AARC-Gains, lower AARC-Losses, more positive ERA, and fewer non-communicable diseases were cross-sectionally associated with greater likelihood of engagement with vigorous physical activity (Adjusted models Odds Ratio (OR) of 1.08; 0.86; and 1.06, respectively). The interactions of AARC-Gains and AARC-Losses with number of non-communicable diseases as cross-sectional predictors of likelihood of engagement with vigorous physical activity were not statistically significant. The interaction between ERA (i.e., General Views on Aging) and number of non-communicable diseases was a statistically significant cross-sectional predictor of likelihood of engagement with vigorous physical activity (OR = 0.99; *p* = 0.044). **Conclusions**: Having more positive and less negative views on aging may prompt vigorous physical activity engagement. Moreover, positive general views of aging may be particularly important for physical activity among those who have one or more non-communicable diseases. Although we cannot infer causality, promoting positive views on aging and decreasing negative views on aging could help fostering active aging, especially among those with physical health conditions.

## 1. Introduction

With population aging occurring worldwide, a growing number of individuals are living with one or more chronic health conditions [1]. Based on the Survey on Health, Aging and Retirement in Europe (SHARE), in 2020 36% of older people (i.e., aged 65 years and over) reported having at least two chronic health conditions. In England, among people aged 65 to 74 years, 73.9% were found to have one or more chronic health conditions and among people aged 75 to 84 years, 86.5% were found to have one or more chronic health conditions [2]. Most of these chronic health conditions are non-communicable diseases, that is, they are not transmitted from one person to another. Non-communicable diseases, such as diabetes, cardiovascular diseases, chronic respiratory disorders, musculoskeletal conditions, and cancer, have a negative impact on individuals’ lives, their families, and the economy of our societies. Finding strategies to manage non-communicable diseases as best as possible is therefore highly important.

In this regard, physical activity is widely recognized as a key component in the management of most non-communicable diseases, with extensive evidence supporting its role in complementing pharmacological treatments, improving quality of life, and reducing mortality risk [3]. Recent evidence highlights that vigorous physical activity, including high-intensity interval training, provides substantial health benefits for older adults with non-communicable diseases. Compared to moderate activity, vigorous exercise leads to greater improvements in glycemic control in type 2 diabetes [4], aerobic capacity in cardiovascular [5] and respiratory diseases [6], joint function in osteoarthritis [7], and reductions in cancer-related fatigue [8]. Moreover, higher-intensity exercise is associated with significant psychological benefits, such as decreased symptoms of depression and anxiety, even in individuals with multiple non-communicable diseases [9].

The World Health Organization’s Guidelines on physical activity and sedentary behavior specifically recommend that individuals living with non-communicable diseases engage in regular physical activity according to their abilities, functional limitations, medication use, and treatment plans [10]. However, differently from the general population, individuals with non-communicable diseases tend to report insufficient levels or lack of physical activity [11,12,13]. In particular, while moderate-intensity activity is more common in older adults [14], vigorous physical activity remains scarce; individuals with non-communicable diseases often perceive vigorous physical activity as risky or unsuitable, potentially leading to under-reporting or avoidance of vigorous physical activity [15,16,17,18]. In England, recent estimates show that only 47.9% of adults living with a chronic health condition or a disability meet the recommended levels of physical activity (i.e., 150 min per week) [19]. Among individuals with chronic health conditions, those with two or more chronic health conditions (i.e., multimorbidity) are at greater risk of being physically inactive [20]. This highlights a critical public health concern, as lower levels of physical activity in this population are associated with worse prognosis and greater disability risk, with meaningful healthcare system cost implications [21,22,23,24,25].

Within the self-care framework, the World Health Organization [26] has emphasized the need for health promotion strategies that effectively support the adoption and maintenance of healthier lifestyle behaviors among older people with non-communicable diseases, including regular physical activity. Such strategies must address not only older adults’ clinical characteristics and functional capabilities but also the psychosocial factors that shape how they perceive their ability to age healthily and engage in health-promoting behaviors. Indeed, these factors may act either as barriers or as motivational levers for sustained physical activity engagement. Among psychological factors, negative views on aging may be a common and relevant barrier towards physical activity engagement that could be addressed through public health and intervention strategies [27]. Views on Aging (VoA) refer to the subjective ways in which individuals perceive, experience, and evaluate their own aging (i.e., personal views on aging; PVoA), as well as the beliefs, expectations, and societal representations they hold about the aging process and older people in general (i.e., general views on aging; GVoA) [28].

According to Stereotype Embodiment Theory [29], stereotypes about old age and aging, internalized from early life, shape how individuals make sense of their aging and influence their behavioral choices. Positive VoA, such as viewing aging as a time of personal growth and new possibilities, can enhance individuals’ sense of agency and control, encouraging them to be proactive in maintaining their health, which in turn leads to better cognitive, physical, and mental health over time [30]. Conversely, negative VoA, such as viewing aging as a time of decline, may reduce motivation and engagement in health-promoting behaviors, and consequently lead to a higher incidence of non-communicable diseases, greater functional decline, increased risk of hospitalization, and higher mortality rates [31]. Among individuals living with one or more non-communicable diseases, negative VoA have been shown to intensify the detrimental impact of such diseases on quality of life [32], predict more severe depressive symptoms [33] and undermine adaptive self-regulation strategies following serious health events [34]. In sum, VoA not only predict risk of new health issues and diseases, but also individuals’ ability to adapt to and cope with health issues and diseases.

Within the VoA literature, physical activity has received some but limited attention, emerging as a behavioral mechanism underlying the association between VoA and health. That is, individuals who associate aging to increased freedom and greater opportunity to invest in self-care (i.e., positive GVoA) are more likely to adopt and sustain a physically active lifestyle [35,36], which can lead to improved health. In contrast, those who view aging primarily in terms of physical decline and energy loss (i.e., negative GVoA) are less likely to engage in regular physical activity [37], potentially increasing their risk for poorer health outcomes. Other studies instead investigated physical activity as an outcome. For example, negative VoA emerged as partial mediators in the relationship between pain and subsequent physical activity. This finding may be due to negative VoA fostering maladaptive interpretations of pain, such as over-generalizing or catastrophizing, which could make people avoid involvement in physical exercise [38]. Although VoA have been extensively linked to health and psychological well-being, their association with levels of physical activity, particularly vigorous physical activity, has been investigated in relatively few studies [35,39]. Studying VoA in relation to this more intense form of physical activity may be especially relevant, as individuals with more negative VoA, who are therefore more likely to hold misconceptions about aging, may also be more prone to endorse the common belief that older people, especially those with one or more health conditions, should avoid intense physical activity due to concerns about physical symptoms such as muscle aches and breathlessness [15]. Moreover, to the best of our knowledge there are no studies investigating whether the interaction between VoA and number of non-communicable diseases explains variance in physical activity levels. This gap is particularly important to address, as identifying psychological factors that help maintain physical activity can inform effective self-care promotion strategies targeting individuals with one or more non-communicable diseases.

This cross-sectional study therefore aims to investigate the associations of PVoA, GVoA, and number of non-communicable diseases with levels of engagement in vigorous physical activity among adults aged 50 and older and living in the United Kingdom. The study also aims to investigate the interactions of PVoA and GVoA with number of non-communicable diseases as cross-sectional predictors of levels of engagement in vigorous physical activity. To assess PVoA, this study uses the Awareness of Age-Related Change 10-item Short-Form Questionnaire (AARC; [40]), a multidimensional instrument that captures both positive (AARC-Gains) and negative (AARC-Losses) perceptions of age-related changes across five domains of life. These are health and physical functioning, cognitive functioning, interpersonal relationships, social–emotional functioning, and lifestyle [41]. To assess GVoA, the study relies on the Expectations Regarding Aging scale (ERA; [42]), which measures individuals’ general beliefs about the aging process in terms of physical health, mental health, and cognitive functioning.

In line with the above reported literature, this study hypothesizes that: (1) higher AARC-Gains and higher ERA will be positively, and higher AARC-Losses will be negatively, associated with vigorous physical activity; (2) A higher number of non-communicable diseases will be associated with lower levels of vigorous physical activity; (3) higher AARC-Gains and more positive ERA will buffer, and AARC-Losses will exacerbate the negative association between a higher number of non-communicable diseases and vigorous physical activity.

## 2. Materials and Methods

This study is based on cross-sectional data collected as part of a broader study investigating VoA in caregivers of people living with dementia in the community in the United Kingdom and in people who never cared for someone with dementia nor for an older person with any other condition. Data was collected online using Qualtrics XM (September 2024–February 2025). The current study sample is based on the subsample of non-caregivers aged 50 years and over. Inclusion criteria for the non-caregivers subsample were: never having cared for someone with dementia nor for an older person with another conditions, being aged 50 or over, speaking English, living in the United Kingdom, and having internet connection. Participants were recruited through advertisements posted on social media, Join Dementia Research, the Enabling Research in Care Homes (ENRICH) network, and snowballing. Because of the nature of recruitment, it was not possible to estimate the response rate. We detected 55 cases of partial attrition, that is 55 individuals started the survey without completing it. Participants were not paid for their participation but were informed that once the target sample size was achieved £2500 would be donated to a UK Charity. This study obtained approval from the Ethics and Governance Committee of the University of Surrey (EGA ref.: FHMS [23,24] 167 EGA). Participants provided their informed consent online prior to completing the survey.

### 2.1. Mesaures

*Personal views on aging* were assessed with the Awareness of Age-Related Change 10-item version [40]. This scale assesses awareness of (perceived) age-related gains and losses (AARC-Gains; AARC-Losses). Respondents rated how much each item applied to them (1 = not at all; 5 = very much). An example of item assessing AARC-Gains is “With my increasing age, I realize that I have more experience and knowledge to evaluate things and people”. An example of item assessing AARC-Losses is “With my increasing age, I realize that I have less energy”. For both the AARC-Gains and AARC-Losses scales, higher scores indicate higher AARC-Gains and AARC-Losses, respectively (range: 5–25). Cronbach’s alphas for the AARC-Gains and AARC-Losses scales were 0.73 and 0.79, respectively.

*General views on aging* were assessed with the 12-item Expectations Regarding Aging (ERA) scale [42]. This scale assesses expectations regarding aging in relation to physical health; mental health; and cognitive function. For each item respondents choose an answer option from “definitely true” (1) to “definitely false” (4). An example of item for the four-item physical health scale is “When people get older, they need to lower their expectations of how healthy they can be”. An example item for the four-item mental health scale is “I expect that as I get older, I will spend less time with friends and family”. An example item for the four-item cognitive function scale is “I expect that as I get older, I will become more forgetful”. Higher scores indicate more positive expectations regarding aging (range: 12–48). Cronbach’s alpha for this scale in the current study sample was 0.83.

*Number of non-communicable diseases* consisted of the sum of the following self-reported conditions (coded 1 as present or 0 as absent): high blood pressure; stroke; heath disease/heart attack/angina; diabetes; mild cognitive impairment; Parkinson’s disease; high cholesterol; hypothyroidism; hyperthyroidism; arthritic condition; Huntington’s disease; current cancer; cancer in full remission; osteoporosis; asthma; epilepsy; motor neurone disease; multiple sclerosis; Paget’s disease; deep vein thrombosis; human immunodeficiency virus; acquired immunodeficiency disease syndrome; and hepatitis C virus.

*Vigorous physical activity* was assessed with a one-item question: “In the last month, how many times have you done any physical activity lasting at least 20 min that has left you out of breath”? Answer options were: 0 times; 1–3 times; 4–10 times; 11–20 times; more than 20 times. This question was taken from the UK PROTECT study (Platform for Research Online to Investigate Genetics and Cognition and Aging) [43].

*Demographic questions* comprised age, sex, ethnicity (White; Non-White), high level of education, marital status, and working status. Ethnicity was used only for descriptive purposes due to the low number (<1%) of non-white participants. Highest level of education comprised the following: No education, primary, or secondary education; post-secondary education; vocational qualification; university degree. Marital status comprised the following: married; widowed; separated or divorced; co-habiting; single. Working status comprised the following: employed full-time; employed part-time; retired; unemployed.

### 2.2. Analyses

Summary statistics of study variables were reported as means with standard deviations (SD) for continuous variables and as a frequency and percent for categorical variables. Pearson’s (for continuous variables) and Spearman’s (for categorical variables) correlation coefficients among study variables were estimated. Univariable and multivariable logistic regression models were estimated to investigate the cross-sectional associations of AARC-Gains, AARC-Losses, and ERA (independent variables) with vigorous physical activity (dependent variable), and of number of non-communicable diseases and likelihood of engagement with vigorous physical activity. Multivariable regression models included age, sex, education, marital status, working status as covariates. The regression models with AARC-Gains, AARC-Losses, and ERA as independent variables also included number of non-communicable diseases as additional covariate. The interactions of AARC-Gains, AARC-Losses, and ERA with number of non-communicable diseases as cross-sectional predictors of likelihood of engagement with vigorous physical activity were also estimated using interaction terms in linear regression models. The level of significance was set at *p* < 0.05. We checked for the assumptions of linear regression models. More specifically, we checked whether vigorous physical activity, the dependent variable in study analyses, was normally distributed by fitting an histogram. We checked for multicollinearity among independent variables and interaction terms by estimating correlation coefficients, with moderate effects suggesting multicollinearity. For correlation coefficients, unstandardized and standardized coefficients were reported. Standardized coefficients ≤ 0.09 were considered negligible; 0.10–0.29 small, 0.30–0.49 moderate, and ≥ 0.50 large [44]. Power calculation using G*Power version 3.1.9.7 suggested that in order to detect a small effect in an interaction model with five covariate and with significance level set at 0.05 and power of 0.80 a sample of 402 individuals is needed. Analyses were conducted in STATA version 18.

## 3. Results

### 3.1. Descriptive Statistics

Participants were 1699 individuals (70.8% women) with a mean age of 67.79 years (SD = 8.05) (Table 1). Almost all (98.1%) were of white ethnicity. Slightly more than half (55.5%) had a university degree; the majority were married (66.5%); and most were retired or unemployed (73.9%). On average participants had one non-communicable disease (range: 0 to 6). Participants on average scored 2.98 on vigorous physical activity indicating that they engaged between four and ten times in vigorous physical activity in the past month. Vigorous physical activity was normally distributed in this study sample. Participants reported more Awareness of Age-Related Gains (AARC-Gains) (M = 19.45) than Awareness of Age-Related Losses (AARC-Losses) (M = 10.44); and an ERA mean score of 29.84 indicating some positive ERA.

### 3.2. Correlations Among Study Variables

A higher number of non-communicable diseases were significantly correlated with older age; lower educational achievements; and being retired (Table 2). Greater engagement in vigorous physical activity was significantly correlated with younger age, highest educational achievement, and fewer non-communicable diseases. Higher AARC-Gains was significantly correlated with younger age, being woman, and greater engagement in vigorous physical activity. Higher AARC-Losses was significantly correlated with older age, being men, lower educational achievement, working, more non-communicable diseases, and less vigorous physical activity. More positive ERA were significantly correlated with younger age, being a women, higher educational achievement, fewer non-communicable diseases, greater engagement in vigorous physical activity, higher AARC-Gains, and lower AARC-Losses. As it can be noted in the correlation matrix, no multicollinearity issues were noted among the independent variables in the linear regression models.

### 3.3. Associations of Personal Views on Aging, General Views on Aging, and Number of Non-Communicable Diseases with Vigorous Physical Activity

In both the unadjusted and adjusted models, higher AARC-Gains were cross-sectionally associated with greater likelihood of engagement with vigorous physical activity (Adjusted model odds ratio, OR = 1.06; 95%CI: 1.04; 1.08; Appendix A). In both the unadjusted and adjusted models, higher AARC-Losses were cross-sectionally associated with less likelihood of engagement with vigorous physical activity (Adjusted model OR = 0.88; 95%CI: 0.86; 0.91; Appendix A). In both the unadjusted and adjusted models, more positive ERA were cross-sectionally associated with greater likelihood of engagement with vigorous physical activity (Adjusted model OR = 1.05; 95%CI: 1.03; 1.07; Appendix A). In both the unadjusted and adjusted models, a higher number of non-communicable diseases was cross-sectionally associated with lower likelihood of engagement in vigorous physical activity (Adjusted model OR = 0.91; 95%CI: 0.85; 0.98; Appendix A).

### 3.4. Interactions of Personal and General Views on Aging with Number of Non-Communicable Diseases as Cross-Sectional Predictors of Vigorous Physical Activity

The interaction between AARC-Losses and number of non-communicable diseases as cross-sectional predictor of vigorous physical activity was not statistically significant in both the unadjusted and adjusted models (Table 3). The interaction between AARC-Gains and number of non-communicable diseases as cross-sectional predictor of likelihood of engagement with vigorous physical activity was almost significant in the unadjusted model (*p* = 0.056) but was not statistically significant in the adjusted model (*p* = 0.77; Table 4). The interaction between ERA and number of non-communicable diseases as cross-sectional predictor of likelihood of engagement with vigorous physical activity was significant both in the unadjusted (OR = 0.99; *p* = 0.027; Table 5) and in the adjusted/multivariable (OR = 0.99; *p* = 0.044) models meaning that the association between having more non-communicable diseases and lower likelihood of engagement with vigorous physical activity is attenuated when individuals have more positive ERA.

## 4. Discussion

This study examined the cross-sectional associations of PVoA, GVoA, and number of non-communicable diseases, with likelihood of engagement with vigorous physical activity among adults aged 50 and over and living in the United Kingdom. Additionally, the study explored whether PVoA and GVoA moderate the relationship between participants’ number of non-communicable diseases and likelihood of engagement with vigorous physical activity. Our findings contribute to the growing body of evidence linking VoA with physical activity [45]. Specifically, in line with our first hypothesis, higher Awareness of Age-Related Gains (AARC-Gains) was associated with greater likelihood of engagement with vigorous physical activity, while higher Awareness of Age-Related Losses (AARC-Losses) was linked to reduced likelihood of engagement with vigorous physical activity levels. Similarly, more positive ERA were associated with greater likelihood of engagement with vigorous physical activity, adding to previous research indicating that positive GVoA are associated with greater likelihood of healthy behavior patterns [46]. Importantly, these associations remained significant in adjusted models, suggesting that VoA have an independent cross-sectional association with vigorous physical activity beyond demographic and health-related variables.

Although the cross-sectional nature of this study does not make it possible to infer causality, VoA may be relevant psychological determinants of health behaviors in middle and later life [27], and could be integrated into behavioral models for middle-aged and older adults as possible antecedents of health behaviors. Moreover, by examining these associations in a sample of individuals with one or more non-communicable diseases, our study suggests that both PVoA and GVoA may be important correlates of physical activity engagement within a population which is often underrepresented in this field. Many cohort studies tend to include individuals with above-average physical health or do not always fully account for the presence and impact of chronic diseases (e.g., [47,48], potentially hiding the specific role that VoA may play among those living with non-communicable diseases. Moreover, the association observed in this study between AARC-Losses, and the number of non-communicable diseases is consistent with previous research reporting more perceived losses among middle-aged and older individuals facing multiple health problems, likely reflecting increased awareness of physical and functional decline. In contrast, AARC-Gains did not show a significant association with the number of non-communicable diseases participants had, further supporting existing literature suggesting that AARC-Gains may be less related to and influenced by health indicators than AARC-Losses [49,50]. However, we found that higher ERA were associated with fewer non-communicable diseases, in line with previous research linking more positive GVoA with more favorable health profiles [46], suggesting that general beliefs about aging may operate independently from individuals’ self-perceptions of aging.

According to our second hypothesis and consistent with previous literature [20], this study found that higher number of non-communicable diseases was associated with lower likelihood of engagement with vigorous physical activity. As individuals grow older and become more likely to experience health-related challenges, their engagement with physical activity might be significantly affected by physical and psychological barriers including functional limitations, fatigue, pain, fear of falling, and often erroneous concerns about worsening their medical conditions [16,51,52]. This is particularly worrisome for people with non-communicable diseases (such as diabetes, cardiovascular diseases, chronic respiratory disorders, musculoskeletal conditions, and cancer), given the essential role of physical activity in preserving physical functioning and preventing complications [21,53].

The study findings also confirmed our third hypothesis that a more positive ERA moderates the association between having a higher number of non-communicable diseases and lower likelihood of engagement with vigorous physical activity. Importantly, the association between having a higher number of non-communicable diseases and lower likelihood of engagement with vigorous physical activity was attenuated when individuals have more positive ERA. Hence, it may be that positive ERA acts as a psychological resource that increased individuals’ likelihood of engaging with vigorous physical activity even when they have one or more non-communicable disease [45,46]. It may be that less fatalistic and deterministic views about cognitive, physical, and emotional aging help individuals perceiving some sort of control over the course of their conditions [54]. That is, individuals with positive VoA may recognize the benefits of remaining physically active, thus sustaining motivation even in the face of physical limitations.

However, contrary to our hypothesis, in the adjusted models, no significant interactions were found between non-communicable diseases and AARC-Gains or AARC-Losses as cross-sectional predictors of likelihood of engagement with vigorous physical activity levels, although trends were observed in the unadjusted analyses. Despite the cross-sectional nature of this data, which prevents from causal interpretation, our findings may suggest that GVoA, more so than PVoA, may play a protective role when facing physical health challenges. PVoA may be more directly influenced by the experience of illness itself [55,56], making them less effective in buffering its association with physical activity. In contrast, GVoA, which are typically created in young age and become consolidated over the life course, may remain more stable and protective in guiding health behaviors even when facing one or more illnesses.

Further research should expand upon these findings, investigating whether and how VoA interact with illness beliefs and representations (i.e., illness perceptions, [57]) in its relationship with health behaviors and outcomes. For instance, when older adults perceive chronic conditions as inevitably linked to aging, they may be less likely to engage in health-promoting behaviors [58]. Similarly, common misconceptions, such as the belief that pain and physical symptoms can be worsened by physical activity, can have a strong deterrent effect on chronic patients’ engagement with physical activity, being also more influential than objective measures of pain themselves [16,17,18]. An improved understanding of the interaction between these beliefs could inform the development of psychoeducational interventions aimed at dispelling existing misconceptions, thereby fostering more accurate and empowering perspectives.

Overall, although due to the cross-sectional nature of this study we cannot infer causality, these findings highlight the potential of targeting VoA in interventions aimed at promoting physical activity among middle-aged and older adults with non-communicable diseases.

According to the World Health Organization self-care framework [26], such interventions require a person-centered approach that considers individuals’ values, perceptions, and life context shaping their health behaviors. In this perspective, promoting physical activity among older people with chronic diseases not only requires a precise assessment of each individual’s health status to ensure that interventions are both clinically safe and feasible. It also might entail addressing their discouraging attitudes and beliefs regarding aging. Reshaping negative VoA, for example, through narrative interventions [59], may help older adults reframe aging not as inevitable decline, but as a life stage where it is still possible to remain active and engaged despite the occurring health concerns. This, in turn, may promote a more proactive approach to health management and self-care. Multicomponent programs [36,60,61], aimed at increasing physical activity through the enhancement of positive VoA, have demonstrated feasibility and efficacy among older adults. Combining physical activity with psychoeducational sessions employing narrative-based or cognitive restructuring approaches has shown promise in increasing physical activity engagement by emphasizing the pleasures, benefits, and rewards of being active, thereby countering dominant narratives of decline [61,62]. Emerging technology-based methods, such as app-delivered self-tracking tools that integrate physical activity monitoring with positive messaging about aging, may also contribute to reshaping maladaptive beliefs by replacing unrealistic expectations with more realistic and balanced perspectives [63]. Such interventions could be tailored for individuals with non-communicable diseases and adapted to address their illness representations, which might intersect with VoA to discourage physical activity. At a broader level, challenging widespread ageist narratives and fostering more positive representations of aging across society are key strategies for promoting healthy aging and self-care. Public health initiatives should emphasize that an active and engaged later life is possible, even in the presence of health challenges [64]. Finally, although not part of our primary study aims, findings related to covariates revealed significant associations with vigorous physical activity engagement. In line with previous research [65,66], this study found that lower educational attainment was associated with lower vigorous physical activity levels. This might reflect reduced health literacy and access to resources, such as exercise facilities and structured programs. Indeed, although income data were not collected in this study, educational level often correlates with socioeconomic status and could indirectly influence opportunities for physical activity [67]. Retirement was also linked to reduced vigorous physical activity engagement. Although it is generally assumed to increase the availability of time for health-promoting activities, retirement can also disrupt daily routines and reduce opportunities for structured physical activity, making engagement more dependent on personal motivation, which may decline in later life [68,69,70,71].

### Strengths and Limitations

This study has several strengths. The first is the assessment of both PVoA and GVoA. This is important as, although related, PVoA and GVoA do not necessarily overlap and they may be differently related to behavioral and health indicators [41]. Second, this is one of the few studies investigating awareness of age-related change as a moderating variable. Finally, the large sample size and consequent power enabled to reliably conduct multiple regression and moderation models. There are also limitations. The cross-sectional nature of the study did not make it possible to investigate causal pathways nor the bidirectional associations of VoA indicators, vigorous physical activity, and non-communicable diseases. Physical activity was assessed using a self-reported single-item question focused on vigorous activity, which may not accurately reflect overall activity levels compared to more comprehensive or objective measures (i.e., accelerometers) and may miss associations with more common moderate-intensity activity. This compromises generalizability. Indeed, differently from when using objective measures, the use of self-reported psychological and behavioral measures can imply some level of social desirability bias. Moreover, for the one item used to assess vigorous physical activity there is no published psychometric information. Future research would benefit from employing multidimensional self-report instruments or objective tools (e.g., accelerometers) to capture a more accurate picture of physical activity patterns. Participants were almost all White so results cannot be extended to other ethnicities. Participants were also all well-educated which limits generalizability to the population of middle-aged and older adults who have a lower educational and socioeconomic background. Women may also be overrepresented in the current study compared to the proportion of older women in the UK population. In addition, participants were recruited online. Hence, those who did not have internet access have been excluded from the study sample, limiting generalizability. By using the number of non-communicable diseases as a variable, our analysis may have missed specific effects associated with particular types of diseases. A final limitation of this study is that the independent variables in the regression model had low explanatory power.

## 5. Conclusions

As physical inactivity and sedentary behavior are major contributors to the global burden of non-communicable diseases, fostering active habits among adults with non-communicable diseases is a public health priority. Findings of this study indicate that participants with more positive and less negative VoA show greater likelihood of engaging in vigorous physical activity and particularly that positive general VoA can moderate the effect of the number of NCDs. As a relevant and modifiable psychological factor, VoA can help align health behavior promotion strategies with the values, needs, and lived experiences of older adults, ultimately enhancing interventions’ effectiveness and contributing to the development of more responsive and equitable care systems. Moreover, countering ageist stereotypes by promoting more positive narratives of aging is essential to support adults in actively managing the burden of chronic conditions. In this context, health-promotion campaigns could challenge discouraging narratives and misconceptions about aging and physical decline, highlighting the possibility of maintaining an active lifestyle even in later life and in spite of chronic diseases.

## Figures and Tables

**Table 1 healthcare-13-02071-t001:** Descriptive statistics of study variables.

Variables	Statistics
N	1699
Age, M (SD; range)	67.79 (8.05)
Sex, n (%)	
Women	1201 (70.8)
Men	495 (29.2)
Ethnic origin, n (%)	
White	1661 (98.1)
Non-white	30 (1.8)
Highest level of education, n (%)	
No education, primary, or secondary education	225 (13.4)
Post-secondary education	192 (11.4)
Vocational qualification	331 (19.7)
University degree	933 (55.5)
Marital status, n (%)	
Married or civil partnership	1118 (66.5)
Widowed	149 (8.9)
Separated/divorced	199 (11.8)
Cohabiting	87 (5.2)
Single	129 (7.7)
Working status, n (%)	
Employed	440 (26.1)
Retired or unemployed	1244 (73.9)
Number of non-communicable diseases, M (SD; range)	1.19 (1.17)
Zero non-communicable diseases, n (%)	601 (35.4)
One non-communicable disease, n (%)	515 (30.3)
Two non-communicable diseases, n (%)	327 (19.3)
Three non-communicable diseases, n (%)	171 (10.1)
Four or more non-communicable diseases, n (%)	85 (5.0)
Physical activity, M (SD)	2.98 (1.29)
Awareness of age-related gains, M (SD)	19.45 (3.80)
Awareness of age-related losses, M (SD)	10.44 (3.38)
Expectations regarding aging, M (SD)	29.84 (5.66)

**Table 2 healthcare-13-02071-t002:** Correlation matrix among study variables.

Variables	1.	2.	3.	4.	5.	6.	7.	8.	9.
	Correlation Coefficients								
1. Age									
2. Sex	−0.09; 0.0003								
3. Highest level of education	0.01; 749	−0.07; 0.0002							
4. Marital status	0.01; 0.697	0.13; 0.001	−0.02; 0.441						
5. Working status	0.52; 0.001	−0.03; 0.234	0.002; 0.921	−0.07; 0.008					
6. Number of non-communicable diseases	0.25; 0.001	−0.02; 0.464	−0.05; 0.036	0.04; 0.098	0.19; 0.001				
7. Vigorous physical activity	−0.05; 0.044	0.04; 0.072	0.09; 0.0003	−0.02; 0.367	−0.01; 0.547	−0.07; 0.003			
8. Awareness of age-related gains	−0.07; 0.003	0.11; 0.001	−0.03; 0.266	−0.01; 0.785	0.004; 0.859	−0.02; 0.383	0.14; 0.001		
9. Awareness of age-related losses	0.19; 0.001	−0.06; 0.008	−0.07; 0.002	0.03; 0.295	0.15; 0.001	0.28; 0.001	−0.23; 0.001	−0.04; 0.136	
10. Expectations regarding aging	−0.08; 0.001	0.15; 0.001	0.12; 0.001	0.05; 0.057	−0.04; 0.071	−0.11; 0.001	0.16; 0.001	0.12; 0.001	−0.49; 0.001

Pearson’s correlation coefficients are reported for continuous variables and Spearman’s correlation coefficients are reported for categorical variables.

**Table 3 healthcare-13-02071-t003:** Interaction of awareness of age-related losses with number of non-communicable diseases as cross-sectional correlate of vigorous physical activity.

	Unadjusted Model		Adjusted Model	
Cross-Sectional Predictors	OR (95% CI)	*p*-Value	OR (95% CI)	*p*-Value
Awareness of age-related losses	0.86 (0.82; 0.89)	0.001	0.86 (0.83; 0.90)	0.001
Number of non-communicable diseases	0.80 (0.63; 1.02)	0.071	0.83 (0.65; 1.05)	0.125
Awareness of age-related losses × number of non-communicable diseases	1.02 (0.99; 1.04)	0.068	1.02 (0.99; 1.04)	0.116
Age			0.99 (0.98; 1.01)	0.411
Sex			1.16 (0.95; 1.41)	0.138
Education				
Primary or secondary education			0.74 (0.56; 0.98)	0.037
Post-secondary education			0.66 (0.50; 0.88)	0.004
Vocational qualification			0.90 (0.72; 1.13)	0.374
University degree			Reference category	
Marital status				
Married or civil partnership			Reference category	
Widowed			1.06 (0.77; 1.47)	0.703
Separated/divorced			1.03 (0.78; 1.36)	0.828
Co-habiting			0.97 (0.66; 1.43)	0.896
Single			0.78 (0.56; 1.07)	0.126
Working status				
Employed			0.85 (0.67; 1.07)	0.169
Retired			Reference category	

OR = Odds ratio.

**Table 4 healthcare-13-02071-t004:** Interaction of awareness of age-related gains with number of non-communicable diseases as cross-sectional correlate of vigorous physical activity.

	Unadjusted Model		Adjusted Model	
Cross-Sectional Predictors	OR (95% CI)	*p*-Value	OR (95% CI)	*p*-Value
Awareness of age-related gains	1.09 (1.06; 1.13)	0.001	1.08 (1.05; 1.12)	0.001
Number of non-communicable diseases	1.26 (0.89; 1.78)	0.197	1.25 (0.88; 1.77)	0.214
Awareness of age-related gains × number of non-communicable diseases	0.98 (0.97; 1.00)	0.056	0.98 (0.97; 1.00)	0.077
Age			0.99 (0.98; 1.01)	0.365
Sex			1.19 (0.98; 1.45)	0.075
Education				
Primary or secondary education			0.68 (0.52; 0.89)	0.006
Post-secondary education			0.62 (0.47; 0.81)	0.001
Vocational qualification			0.90 (0.72; 1.12)	0.339
University degree			Reference category	
Marital status				
Married or civil partnership			Reference category	
Widowed			0.93 (0.67; 1.28)	0.653
Separated/divorced			1.00 (0.75, 1.31)	0.974
Co-habiting			1.05 (0.71; 1.53)	0.819
Single			0.83 (0.60; 1.14)	0.248
Working status				
Employed			0.95 (0.75; 1.19)	0.633
Retired			Reference category	

OR = Odds Ratio.

**Table 5 healthcare-13-02071-t005:** Interaction of expectations regarding aging with number of non-communicable diseases as cross-sectional correlate of vigorous physical activity.

	Unadjusted Model		Adjusted Model	
Cross-Sectional Predictors	OR (95% CI)	*p*-Value	OR (95% CI)	*p*-Value
Expectations regarding aging	1.07 (1.05; 1.09)	0.001	1.06 (1.04; 1.08)	0.001
Number of physical health conditions	1.40 (0.96; 2.05)	0.083	1.38 (0.94; 2.03)	0.105
Expectations regarding aging × number of non-communicable diseases	0.99 (0.97; 0.99)	0.027	0.99 (0.97; 0.99)	0.044
Age			0.99 (0.98; 1.00)	0.187
Sex			1.13 (0.93; 1.38)	0.220
Education				
Primary or secondary education			0.76 (0.57; 0.99)	0.047
Post-secondary education			0.66 (0.50; 0.88)	0.004
Vocational qualification			0.94 (0.75; 1.18)	0.618
University degree			Reference category	
Marital status				
Married or civil partnership			Reference category	
Widowed			0.96 (0.70; 1.33)	0.816
Separated/divorced			0.99 (0.75; 1.31)	0.952
Co-habiting			1.00 (0.68; 1.47)	0.995
Single			0.74 (0.54; 1.02)	0.070
Working status				
Employed			0.91 (0.72; 1.15)	0.430
Retired			Reference category	

OR = Odds Ratio.

## Data Availability

Study data will be deposited in Open Science Framework and made freely accessible in April 2026.

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
