# Peer review of "Personal and General Views on Aging, Non-Communicable Diseases, and Their Interaction as Cross-Sectional Correlates of Vigorous Physical Activity in UK Individuals Aged 50+"

_healthcare, 2025, doi:10.3390/healthcare13162071_

Round 1
Reviewer 1 Report
Comments and Suggestions for Authors
REVIEW REPORT (healthcare-3761046)
Personal and General Views on Aging, Non-Communicable Diseases, and their interaction as cross-sectional correlates of vigorous physical activity in 50+ UK individuals
ABSTRACT
-The abstract is well written. I would just like to suggest that you confirm, in the text itself, that the association between Views on Aging (VoA) and Vigorous Physical Activity (VPA) remained after adjusting for age, sex, education, etc. Furthermore, I recommend that you clarify that the significant interaction effect was observed only for General Views on Aging (GVoA).
INTRODUCTION
-I noticed that the relationship between VoA and physical health appears several times, such as in lines 83–86 and again between lines 94–102, while the Stereotype Embodiment theory is introduced late. I suggest you summarize the information about this theorie when presenting the VoA construct, eliminating conceptual repetition and making the text more straightforward.
-I also noticed that the hypotheses are diluted at the end of the introduction, within a running paragraph. This makes it difficult to quickly and clearly identify the propositions tested in the study. Please highlight them.
- Although the relationship between NCDs and physical activity is well contextualized, it lacks information on the specific role of vigorous physical activity (VPA) in mitigating the effects of multimorbidity. Please review thispoint.
METHODS
- I would like to know if there is psychometric information on the VPA assessment used. If not, I suggest this limitation be disclosed and discussed in the study.
- While I acknowledge the large sample size, it is important to remember that statistical power depends on the size of the effect to be detected. This is important in the case of the interactions tested, such as between VoA and the number of NCDs on VPA, since the power to detect interactions is generally lower than for main effects. I wonder if it would be possible to include some information on the specific power calculation for these interactions, which could help clarify whether the non-significant effects are in fact due to the absence of an effect or to limited statistical power.
- In addition, was weighting by sample weights used in the analyses? This information is relevant for interpreting the findings.
- General comments: The authors need to includet the inclusion criteria, as they only mention "non-caregivers aged 50+ with internet access," and if possible, add information on response rates, attrition, or exclusions.
RESULTS
- I did not identify excessive homogeneity in the datas, as the standard deviations presented seem plausible. Furthermore, although the effects found are modest, they are also consistent with what is expected in the literature for psychosocial predictors of health behavior.
DISCUSSION
- Please discuss the limitation of the outcome (self-reported and unidimensional VPA), considering that the lack of assessment of moderate PA compromises generalizability.
- How can we intervene in VPA in practice? Whats strategies would be recommended for older adults with NCDs? I suggest revising the writing in this section to emphasize the clinical and public health implications, including concrete examples of VoA-based interventions, such as narratives, cognitive restructuring, or multicomponent programs.
- In the limitations section, discuss internet selection bias, the risks of social desirability bias in psychological and health behavior instruments, and the low explanatory power of the models (R² between 0.01 and 0.04).
CONCLUSION
- Please respond directly to the hypotheses and includes indications of how the findings can guide health promotion policies for older adults with NCDs.
Author Response
ABSTRACT
-The abstract is well written. I would just like to suggest that you confirm, in the text itself, that the association between Views on Aging (VoA) and Vigorous Physical Activity (VPA) remained after adjusting for age, sex, education, etc. Furthermore, I recommend that you clarify that the significant interaction effect was observed only for General Views on Aging (GVoA).
We have implemented the suggested changes in the abstract.
INTRODUCTION
-I noticed that the relationship between VoA and physical health appears several times, such as in lines 83–86 and again between lines 94–102, while the Stereotype Embodiment theory is introduced late. I suggest you summarize the information about this theorie when presenting the VoA construct, eliminating conceptual repetition and making the text more straightforward.
As suggested, we now introduce Stereotype Embodiment Theory earlier in the introduction and we have merged the sentences referring to the associations of VoA with health.
-I also noticed that the hypotheses are diluted at the end of the introduction, within a running paragraph. This makes it difficult to quickly and clearly identify the propositions tested in the study. Please highlight them.
We have now separated the hypotheses from the remaining of the paragraph to make them more easily identifiable.
- Although the relationship between NCDs and physical activity is well contextualized, it lacks information on the specific role of vigorous physical activity (VPA) in mitigating the effects of multimorbidity. Please review thispoint.
In the introduction section of the manuscript, we have added paragraphs highlighting the specific role of vogorous physical activity.
METHODS
- I would like to know if there is psychometric information on the VPA assessment used. If not, I suggest this limitation be disclosed and discussed in the study.
The question we used to assess VPA is a bespoken question that was used in the large UK PROTECT study. We are not aware of published psychometric information on this question. Hence, we mention this as a limitation of the current study.
- While I acknowledge the large sample size, it is important to remember that statistical power depends on the size of the effect to be detected. This is important in the case of the interactions tested, such as between VoA and the number of NCDs on VPA, since the power to detect interactions is generally lower than for main effects. I wonder if it would be possible to include some information on the specific power calculation for these interactions, which could help clarify whether the non-significant effects are in fact due to the absence of an effect or to limited statistical power.
We have run a power analysis. According to G*Power the required sample size to detect a small effect with a moderation model with five covariates is 402 participants. Hence, the current study sample of 1699 should be sufficient to detect even a small significant interaction. The calculation has been for a regression model with eight predictors (i.e., 5 covariates + main predictor + moderator + interaction between predictor and moderator) with a significance level set at 0.05 and power set at 0.80. We now report details of power calculation in the analyses section of the method.
- In addition, was weighting by sample weights used in the analyses? This information is relevant for interpreting the findings.
No, weighting by sample weights was not used in the analyses. We have compared the current study sample with available data for the UK population (2021 data from the Office of National Statistics). From our observation, it seems women may be overrepresented in the current study. We have therefore now added this as a limitation of the current study.
- General comments: The authors need to includet the inclusion criteria, as they only mention "non-caregivers aged 50+ with internet access," and if possible, add information on response rates, attrition, or exclusions.
We have added more details on inclusion criteria in the materials and methods section. It is difficult/impossible to estimate the response rate for this study as participants were not directly invited to take part in the study via email, but through dissemination of study adverts via social media and through third parties (e.g., Join Dementia Research notifying subscribers about the possibility to participate in a study). There were 55 cases of partial attrition i.e., individuals started the survey without completing it. We now mention this in the methods section. There were no cases of excluion for the non-caregivers sample, that is the sample used in this study.
RESULTS
- I did not identify excessive homogeneity in the datas, as the standard deviations presented seem plausible. Furthermore, although the effects found are modest, they are also consistent with what is expected in the literature for psychosocial predictors of health behavior.
If we understand corrrectly, this comment does not imply/require any changes.
DISCUSSION
- Please discuss the limitation of the outcome (self-reported and unidimensional VPA), considering that the lack of assessment of moderate PA compromises generalizability.
We have better highlighted this in the limitation section.
- How can we intervene in VPA in practice? Whats strategies would be recommended for older adults with NCDs? I suggest revising the writing in this section to emphasize the clinical and public health implications, including concrete examples of VoA-based interventions, such as narratives, cognitive restructuring, or multicomponent programs.
We have elaborated on these points in the discussion section.
- In the limitations section, discuss internet selection bias, the risks of social desirability bias in psychological and health behavior instruments, and the low explanatory power of the models (R² between 0.01 and 0.04).
The limitations section already mentions internet selection bias, but we have now also added the risk of social desirability and the low explanatory power of the models as additional limitations of the study.
CONCLUSION
- Please respond directly to the hypotheses and includes indications of how the findings can guide health promotion policies for older adults with NCDs.
We have added indications of how the findings can guide health promotion policies for older adults with NCDs.
Reviewer 2 Report
Comments and Suggestions for Authors
Revision of the Scientific Article: Personal and General Views on Aging, Non-Communicable Diseases, and their interaction as cross-sectional correlates of vigorous physical activity in 50+ UK individuals.
This manuscript addresses a timely and socially relevant topic by investigating the associations between personal and general views on aging, non-communicable diseases (NCDs), and engagement in vigorous physical activity among individuals aged 50 years and older in the UK. The theoretical grounding is sound, the sample is large and well-characterized, and the analytical methods are appropriate for the study’s objectives. The study adds meaningful insight to the literature, particularly in understanding psychological determinants of health behavior in populations with chronic health conditions. The manuscript is well-written, and the structure adheres to scientific standards. However, there are areas that would benefit from clarification or additional elaboration. These are outlined below.
Comments
- Cross-Sectional Design and Interpretation of Findings: Although the authors acknowledge the cross-sectional nature of the study, further emphasis is needed when discussing the limitations this imposes on causal inference. In some sections of the Discussion, causal language ("promoting", "may increase") could be misinterpreted.
- Measurement of Physical Activity: The use of a single self-reported item focusing exclusively on vigorous physical activity may limit the comprehensiveness of the findings. The rationale for excluding moderate-intensity activity, which is more common among older adults, should be better justified, or at least discussed as a limitation.
- Sample Representativeness: The sample is largely composed of white, well-educated participants recruited online. While this is noted in the Limitations section, the authors should explicitly address how this affects generalizability, especially to more socioeconomically and ethnically diverse populations.
- Terminology Clarification: The acronyms AARC-Gains and AARC-Losses are used frequently; a brief reminder of their meanings when reintroduced in the Results and Discussion sections would aid readability.
- Tables and Supplementary Materials: Tables are clear and well-labeled. However, references to supplementary tables should indicate where they can be accessed, especially prior to formal publication.
- Reference Formatting: Ensure consistency in reference formatting (e.g., punctuation, journal names). A few minor inconsistencies were noted in the reference list. It would be good to review it
The manuscript is of high quality and merits publication after minor revisions. Addressing the points above—particularly expanding on the limitations regarding study design and measurement—will strengthen the overall contribution and clarity of the work.
Author Response
Comments
- Cross-Sectional Design and Interpretation of Findings: Although the authors acknowledge the cross-sectional nature of the study, further emphasis is needed when discussing the limitations this imposes on causal inference. In some sections of the Discussion, causal language ("promoting", "may increase") could be misinterpreted.
We have emphasised this in the limitations section and revised/toned down the language throughout the discussion.
- Measurement of Physical Activity: The use of a single self-reported item focusing exclusively on vigorous physical activity may limit the comprehensiveness of the findings. The rationale for excluding moderate-intensity activity, which is more common among older adults, should be better justified, or at least discussed as a limitation.
We mention this in the limitations section and in the revised version of the manuscript we have emphasised this limitation even more. In the introduction of the manuscript we have added paragraphs enphasising the unique role of vigorous physical activity.
- Sample Representativeness: The sample is largely composed of white, well-educated participants recruited online. While this is noted in the Limitations section, the authors should explicitly address how this affects generalizability, especially to more socioeconomically and ethnically diverse populations.
The race limitation and its consequences was already explained in the limitations section of the discussion. We have now added the inclusion of well-educated individuals as a limitation of the current study.
- Terminology Clarification: The acronyms AARC-Gains and AARC-Losses are used frequently; a brief reminder of their meanings when reintroduced in the Results and Discussion sections would aid readability.
Comment implemented.
- Tables and Supplementary Materials: Tables are clear and well-labeled. However, references to supplementary tables should indicate where they can be accessed, especially prior to formal publication.
At this stage of the review we make reference to supplementary tables in the text of the manuscript. We asume that, should the paper be accepted for publication, the editoral office will add more explicit reference to supplementary material.
- Reference Formatting: Ensure consistency in reference formatting (e.g., punctuation, journal names). A few minor inconsistencies were noted in the reference list. It would be good to review it
We have revised the references and applied changes to ensure consistence in the formatting.
The manuscript is of high quality and merits publication after minor revisions. Addressing the points above—particularly expanding on the limitations regarding study design and measurement—will strengthen the overall contribution and clarity of the work.
Thank you. Aligned with the comments of this and the remaining reviewers we have substantially expanded the limitations section.
Reviewer 3 Report
Comments and Suggestions for Authors
The article examines the relationship between individual and general perceptions of aging (AARC-Gains, AARC-Losses, ERA) and the number of chronic diseases and vigorous physical activity. It is an important article for public health.
The article contains some omissions. It can be published after correction.
1) What was the sample size based on?
What are Type I and Type II errors? If a power analysis was not performed at the beginning of the study, a post-hoc power analysis is recommended.
2) Whether the variables exhibited a normal distribution was analyzed. This information was not provided in the statistical analysis section.
3) In the regression model, linear regression was applied to ordinal variables; a logistic regression model was more appropriate. Only correlation analysis was performed for pairwise comparisons, and linear regression analysis was then performed. In pairwise comparisons, current and logistic regression analysis should be applied to variables suitable for chi-square analysis.
4) "Physical activity was almost significant in the unadjusted model (p= .051; Table 3)" was written. This problem can be resolved by entering a significance level of p<=0.05 in the statistical analysis section.
5) Why was vigorous physical activity included, and was it assumed that moderate physical activity had no effect?
6) Multicollinearity control was not performed in the interactions. The multicollinearity control should be explained. For example, BMI = kg/height (m²). Both weight and height should not be included in the model; only BMI values should be used.
Author Response
The article examines the relationship between individual and general perceptions of aging (AARC-Gains, AARC-Losses, ERA) and the number of chronic diseases and vigorous physical activity. It is an important article for public health.
The article contains some omissions. It can be published after correction.
1) What was the sample size based on?
What are Type I and Type II errors? If a power analysis was not performed at the beginning of the study, a post-hoc power analysis is recommended.
This study uses data for the subsample of non-caregivers collected as part of a broader sample/study aiming to compare views on aging between dementia caregivers and non-caregivers. For the purpose of the original study we did an a-priori sample size calculation and we estimated that we needed at least 170 non-caregivers. However, the current study implied additional analyses, including moderation models. We have therefor conducted a post-doc power analysis and found that we need 402 participants to detect a small effect with significance level set al 0.05 and power at 0.80. We have reported details of this analyses in the analyses section of the method.
2) Whether the variables exhibited a normal distribution was analyzed. This information was not provided in the statistical analysis section.
For the regression models, we analysed whether the outcome/dependent variable (physical activity) was normally distributed by fitting an histogram. The variable was normally distributed. That is also why we treated it as a continuous variable in the analyses. We have added detailed in the analyses and results sections as requested.
3) In the regression model, linear regression was applied to ordinal variables; a logistic regression model was more appropriate. Only correlation analysis was performed for pairwise comparisons, and linear regression analysis was then performed. In pairwise comparisons, current and logistic regression analysis should be applied to variables suitable for chi-square analysis.
We used linear regression models because the outcome (physical activity) showed a normal distribution (please see the below histogram) as hence we treated it as a continuous variable. Nonetheless, we have now run logistic regressions and updated study results accordingly.
4) "Physical activity was almost significant in the unadjusted model (p= .051; Table 3)" was written. This problem can be resolved by entering a significance level of p<=0.05 in the statistical analysis section.
Comment implemented.
5) Why was vigorous physical activity included, and was it assumed that moderate physical activity had no effect?
We have added several paragraphs to the introduction section to explain why we specifically focused on vigorous physical activity.
6) Multicollinearity control was not performed in the interactions. The multicollinearity control should be explained. For example, BMI = kg/height (m²). Both weight and height should not be included in the model; only BMI values should be used.
In Table 2 we report the correlations of AARC-gains, AARC-losses, and ERA with number of non-communicable diseases. As it can be seen, the associations are either non-significant or of small size, which allows us to exclude multicollinearity for the interaction. For what regards the covariates age, sex, education, marital status, and working status, these also show either non-significant or small correlations with AARC-gains, AARC-losses, and ERA. This again suggests there is no issue of multicollinearity in the model. We now discuss multicollinearity in the analyses and results section so this reasoning is explicit to the reader.
Round 2
Reviewer 1 Report
Comments and Suggestions for Authors
None
Author Response
I believe the reviewer did not suggest any further changes. Thank you.
Reviewer 3 Report
Comments and Suggestions for Authors
Corrections made by the author are acceptable.
Author Response
The reviewer was satisfied with the changed we have made to the paper.